# An Overview on Phenotypic and Genotypic Characterisation of Carbapenem-Resistant *Enterobacterales*

**DOI:** 10.3390/medicina58111675

**Published:** 2022-11-19

**Authors:** Ali A. Rabaan, Khalid Eljaaly, Saad Alhumaid, Hawra Albayat, Wasl Al-Adsani, Amal A. Sabour, Maha A. Alshiekheid, Jumana M. Al-Jishi, Faryal Khamis, Sara Alwarthan, Mashael Alhajri, Amal H. Alfaraj, Huseyin Tombuloglu, Mohammed Garout, Duaa M. Alabdullah, Elmoeiz Ali Elnagi Mohammed, Fatimah S. Al Yami, Haifa A. Almuhtaresh, Kovy Arteaga Livias, Abbas Al Mutair, Shawqi A. Almushrif, Mai Abdel Haleem A. Abusalah, Naveed Ahmed

**Affiliations:** 1Molecular Diagnostic Laboratory, Johns Hopkins Aramco Healthcare, Dhahran 31311, Saudi Arabia; 2College of Medicine, Alfaisal University, Riyadh 11533, Saudi Arabia; 3Department of Public Health and Nutrition, The University of Haripur, Haripur 22610, Pakistan; 4Department of Pharmacy Practice, Faculty of Pharmacy, King Abdulaziz University, Jeddah 21589, Saudi Arabia; 5Pharmacy Practice and Science Department, College of Pharmacy, University of Arizona, Tucson, AZ 85716, USA; 6Administration of Pharmaceutical Care, Al-Ahsa Health Cluster, Ministry of Health, Al-Ahsa 31982, Saudi Arabia; 7Infectious Disease Department, King Saud Medical City, Riyadh 7790, Saudi Arabia; 8Department of Medicine, Infectious Diseases Hospital, Kuwait City 63537, Kuwait; 9Department of Infectious Diseases, Hampton Veterans Administration Medical Center, Hampton, VA 23667, USA; 10Department of Botany and Microbiology, College of Science, King Saud University, Riyadh 11451, Saudi Arabia; 11Internal Medicine Department, Qatif Central Hospital, Qatif 635342, Saudi Arabia; 12Infection Diseases Unit, Department of Internal Medicine, Royal Hospital, Muscat 1331, Oman; 13Department of Internal Medicine, College of Medicine, Imam Abdulrahman Bin Faisal University, Ammam 34212, Saudi Arabia; 14Pediatric Department, Abqaiq General Hospital, First Eastern Health Cluster, Abqaiq 33261, Saudi Arabia; 15Department of Genetics Research, Institute for Research and Medical Consultations (IRMC), Imam Abdulrahman Bin Faisal University, Dammam 34221, Saudi Arabia; 16Department of Community Medicine and Health Care for Pilgrims, Faculty of Medicine, Umm Al-Qura University, Makkah 21955, Saudi Arabia; 17Molecular Diagnostic Laboratory, Dammam Regional Laboratory and Blood Bank, Dammam 31411, Saudi Arabia; 18Department of Clinical Laboratory Sciences, Prince Sultan Military College of Health Sciences, Dhahran 34313, Saudi Arabia; 19Department of Medical Laboratory, King Fahad Military Medical Complex, Dhahran 34313, Saudi Arabia; 20Department of Clinical Laboratories Services, Dammam Medical Complex, Dammam Health Network, Dammam 5343, Saudi Arabia; 21Facultad de Ciencias de la Salud, Universidad Científica del Sur, Lima 15001, Peru; 22Facultad de Medicina, Universidad Nacional Hermilio Valdizán, Huánuco 10000, Peru; 23Research Center, Almoosa Specialist Hospital, Al-Ahsa 36342, Saudi Arabia; 24College of Nursing, Princess Norah Bint Abdulrahman University, Riyadh 11564, Saudi Arabia; 25School of Nursing, Wollongong University, Wollongong, NSW 2522, Australia; 26Nursing Department, Prince Sultan Military College of Health Sciences, Dhahran 33048, Saudi Arabia; 27Department of Microbiology and Hematology Laboratory, Dammam Comprehensive Screening Centre, Dammam 31433, Saudi Arabia; 28Faculty of Medical Allied Science, Zarqa University, Zarqa 13110, Jordan; 29Department of Medical Microbiology and Parasitology, School of Medical Sciences, Universiti Sains Malaysia, Kubang Kerian 16150, Malaysia

**Keywords:** Carbapenem, multi-drug resistance, MDR, β-lactamase, carbapenemase, antibiotic resistance, *Enterobacterales*, *Enterobacteriaceae*

## Abstract

Improper use of antimicrobials has resulted in the emergence of antimicrobial resistance (AMR), including multi-drug resistance (MDR) among bacteria. Recently, a sudden increase in Carbapenem-resistant *Enterobacterales* (CRE) has been observed. This presents a substantial challenge in the treatment of CRE-infected individuals. Bacterial plasmids include the genes for carbapenem resistance, which can also spread to other bacteria to make them resistant. The incidence of CRE is rising significantly despite the efforts of health authorities, clinicians, and scientists. Many genotypic and phenotypic techniques are available to identify CRE. However, effective identification requires the integration of two or more methods. Whole genome sequencing (WGS), an advanced molecular approach, helps identify new strains of CRE and screening of the patient population; however, WGS is challenging to apply in clinical settings due to the complexity and high expense involved with this technique. The current review highlights the molecular mechanism of development of Carbapenem resistance, the epidemiology of CRE infections, spread of CRE, treatment options, and the phenotypic/genotypic characterisation of CRE. The potential of microorganisms to acquire resistance against Carbapenems remains high, which can lead to even more susceptible drugs such as colistin and polymyxins. Hence, the current study recommends running the antibiotic stewardship programs at an institutional level to control the use of antibiotics and to reduce the spread of CRE worldwide.

## 1. Introduction

One of the primary reasons for the emergence of antibiotic resistance (AMR) worldwide is over-the-counter availability of antibiotics. With the incidences rising alarmingly, AMR poses severe challenges to the general public and the medical fraternity. AMR accounts for a significant proportion of the global morbidity and mortality rates associated with bacterial infections. The most important contributor to multi-drug resistance (MDR) is Gram-negative bacteria. Recently, MDR focus has been placed on Carbapenem-resistant Gram-negative bacteria. The World Health Organization (WHO) lists CRE, Carbapenem-resistant *Acinetobacter baumannii* (CRAB), and Carbapenem-resistant *Pseudomonas aeruginosa* (CRPA) as priority AMR pathogens that pose significant threats to human health [1].

*Enterobacterales* are Gram-negative facultative anaerobes that cause a broad spectrum of severe infections such as septicemia, community/hospital/ventilator-acquired infections, complex urinary tract infections (cUTIs), and intra-abdominal infections [2,3,4]. Because of the wide range of infections caused by this group of bacteria, AMR due to these bacteria has a significant effect at the socio-economic and public health levels. Carbapenem is widely used to treat infections caused by extended-spectrum b-lactamase (ESBL)-producing *Enterobacterales*. Widespread use of Carbapenem against these organisms has led to the emergence of carbapenemase-producing *Enterobacterales* (CPE), whose infections are challenging to treat with the other drug [5]. Carbapenem acts by inhibiting peptidoglycan synthesis by inhibiting the transpeptidases. The CRE develops resistance to Carbapenem by either producing carbapenemase enzyme that digests the Carbapenem or the acquired structural mutations that induce the production of other β-lactamases [6].

There are very few treatment options for CRE infections, making this issue even worse. With an increasing incidence of CRE infections and lack of treatment modalities for CRE infections, it becomes imperative to analyse the molecular mechanisms of MDR developed by CRE. This review focus is to discuss the prevalence of CRE, the molecular mechanism of MDR developed by CRE, modes of transmission of CRE, the genetics involved, diagnostics, and treatment modalities available to manage CRE infections.

## 2. Development of Carbapenem Resistance: Molecular Mechanisms

Carbapenem is an antibiotic against severe bacterial infections, although often reserved for MDR bacterial infections. It is a β lactam antibiotic that works like penicillin and cephalosporins, which belong to the same class. Carbapenem exerts its antibiotic effects by inhibiting transpeptidases, thereby inhibiting peptidoglycan synthesis. Inhibition of peptidoglycan synthesis in the Gram-negative bacteria causes the cells to undergo lysis [6]. The Gram-negative CRE has developed resistance to the carbapenems. The resistance offered by CREs against Carbapenem is usually caused by either hydrolysis of the antibiotic by carbapenemase they produce or structural mutations that induce expression of other β-lactamases including *AmpC* cephalosporinase and ESBL [7,8]. Carbapenemase are usually categorised into one of three classes: Class-A *Klebsiella pneumoniae* carbapenemase (KPC), Class-B metallo-β-lactamases (MBLs), and Class-D oxacillinases (OXA)-type enzymes. The enzymes from different classes exhibit differential inhibitory effects on carbapenem. The beta-lactamases are further classified into 4 classes (Figure 1).

Class B carbapenemases, also called Metallo beta-lactamases (MBLs), have at least one zinc ion to break down the beta-lactam ring. Class A and D carbapenemases have a serine amino acid at their active site. *Klebsiella pneumoniae* carbapenemase (KPC) is the most frequently detected Class A carbapenemase, and it is currently the most common in the United States [6]. Penicillins, cephalosporins, carbapenems, and aztreonam (ATM) may all be hydrolyzed by them, and clavulanic acid can partly block them. Although they may also be found in non-fermenters Gram-negative bacteria, they are more often seen in *Enterobacterales*. This category includes carbapenemases such as *Serratia marcescens* enzyme (SME), imipenem-hydrolyzing beta-lactamase, non-metallo-carbapenemase of class A (NMC-A), and Guyana extended-spectrum beta-lactamase (GES). But aztreonam and ion chelators such as ethylenediaminetetraacetic acid (EDTA) or dipicolinic acid (DPA) block class B carbapenemases, which helps treat or identify these microorganisms [7].

Additionally, when no additional resistance mechanisms are exhibited, MBLs are resistant to beta-lactamase inhibitors but remain susceptible to ATM. The Verona integron-encoded MBL (VIM), the New Delhi MBL (NDM), and imipenemase are some of the MBLs with the greatest clinical significance. Due to the many sequence variants that have been discovered within this group, class D carbapenemases (sometimes referred to as oxacillinases or OXAs) comprise the largest class of carbapenemases. Class D carbapenemases differ from class A and class B in that they are resistant to beta-lactamase inhibitors and EDTA, have only modest hydrolytic activity against carbapenemases, and have no impact on extended-spectrum cephalosporins. The majority of Class D carbapenemases have been discovered in *Acinetobacter* spp., in particular, OXA-51, which is chromosomally expressed in *Acinetobacter baumannii* and may confer carbapenem resistance. Notably, although they are found in *Acinetobacter* spp., OXA-48-like enzymes are the only subgroup with an actual prevalence in the *Enterobacterales* [6,8].

## 3. Treatment Modalities

The growing use of carbapenem has caused more selection for CRE genes, consequently accelerating the emergence of CRE [9]. The rise in the prevalence of CRE worldwide, with limited treatment options available, has made the situation worse. Polymyxins (colistin or polymyxin B) and tigecycline have been the drugs of choice since the beginning to treat CRE infections [10]. However, resistance against these antibiotics is increasing steadily. Fosfomycin and aminoglycosides are the other antibiotics used occasionally to treat CRE infections [3,4,11]. When treating CRE infections of lower minimum inhibitory concentration (MIC), carbapenems remain the treatment of choice but have to be used in high doses or as part of combined-drug therapy.

### Limited Treatment Options

However, there are concerns about the effectiveness, adverse effects, and increasing resistance against these antibiotics [12]. Some CRE-infected patients have shown resistance to colistin [13]. The colistin-resistant genes (*mcr 1-5*) carried on the plasmids are transferable between bacteria, thereby spreading the colistin resistance [14,15,16,17,18,19]. The European Medicines Agency recently authorised the use of a new antibiotic combination (ceftazidime-avibactam) in treating complicated infections, HAP, and those resulting from aerobic Gram-negative bacteria [3]. There is inadequate confirmation of the results, and there are high chances of developing resistance [3].

Combination therapy serves the purpose in treating several complex infectious diseases. Combination therapy has been shown to increase the efficacy of treatment, but it also increases the side effects as well as the probability of death [20,21]. Specific combination therapies have been approved, including meropenem-vaborbactam and imipenem-relebactam, while others are still in development, such as aztreonam-avibactam [3]. There are additional agents approved recently and used for CRE, such as plazomicin, cefiderocol, and eravacycline [3,20]. Therefore, for CRE infections, combination therapies/combinations of antimicrobial agents were tested by several groups both in vivo and in vitro and have been shown to have survival benefits [21,22]. However, these studies do not provide reliable evidence because of variations in research design as well as mechanisms of resistance. Figure 2 summarises the therapeutic options for CRE infections.

## 4. Epidemiology of Carbapenem Resistance

The carbapenem resistance is carried out by the genes present on the plasmids of the bacteria. These genes encode for β-lactamases [23]. The carbapenemase enzymes hydrolyse all β-lactam antibiotics, thereby conferring carbapenem resistance to the bacteria. The *Enterobacterales* carry the resistance genes on their plasmids that express the most important carbapenemases, i.e., KPC and the New Delhi Metallo-β-lactamase (NDM). KPC has been widely reported in the United States, southern Europe, Israel, and China, while NDM has been reported in northern Europe, the UK, and India [24,25,26]. KPC is the commonest of all the carbapenemases globally, followed by NDM [27]. The third most common carbapenemase worldwide is the OXA-48-type oxacillinase. It has been reported mostly in North Africa and Europe. Verona integron-encoded and imipenemase metallo-β-lactamases (VIM and IMP) are the two other metallo-β-lactamase carbapenemase like NDM. They share a similar mechanism of transmission of resistance, although they are rarer.

From a U.S. study conducted in 2016, 48 states reported the presence of CRE resulting from KPC, 25 states reported NDM, 19 states reported OXA-48, and 6 states reported VIM [28]. Currently, there is no efficient treatment for CRE as they are resistant to almost all the antibiotics available. Colistin is one of the antibiotics used against CRE, but recent reports have documented colistin resistance among CRE [29,30]. The very first report of colistin-resistant CRE came from the U.S. The mcr-1 gene present conferred the resistance on the plasmid [31].

## 5. Prevalence of CRE

The reports of the prevalence of CRE infections come from studies conducted in different countries. In an extensive study involving CRE cases from 7 states in the U.S., CRE incidences were reported to be 2.93 per every 100,000 individuals in the U.S. [32]. Despite prompt action and in-progress control measures, CRE have spread to other states such as Orange County, California, and Chicago, Illinois [33,34]. Since the 2000s, the prevalence of CRE has been quite high in New York [35]. The 2013 CDC Antibiotic Resistance Threat Report highlighted that every year there are 9000 new healthcare-associated CRE infections in the U.S. that result in mortality rates of 6.6% [36]. A recent systematic review on the prevalence of CRE in the U.S. reported a prevalence range of 0.3–2.93 infections per 100,000 person-years throughout the U.S. population [37].

A study conducted in a large university hospital in Bangkok, Thailand, reported a CRE prevalence of 1.4% between 2009 and 2011 [38]. In a recent study conducted in Thailand, the quarterly CRE incidence was reported to be significantly increased from 3.37 per 100,000 patient-days in 2011 to 32.49 per 100,000 patient-days in 2016 [39]. CRE have been identified from almost all the regions of China [40,41]. A recent study reported the CRE incidence rate of 4.0 per 10,000 discharges among 25 tertiary hospitals in China [42].

## 6. Transmission of CRE

### 6.1. Healthcare Settings

CRE have significant potential to result in hospital outbreaks; some of which have been reported in hospitals in European countries [43,44,45,46,47,48]. Several studies have investigated potential risk factors that could predispose hospitals to such outbreaks. Several days in the intensive care unit (ICU), critical illnesses, use of invasive devices, history of antimicrobial therapy, as well as long-term in-hospital care are some of the reported risk factors for CRE spread in hospitals [49,50,51,52]. Evidence from a recent, extensive, systematic review pointed toward widespread use of carbapenem and invasive medical devices as the biggest contributors to CRE acquirement by patients [53]. Carbapenemase genes present on the plasmids that confer the resistance against carbapenem have the tendency to get exchanged between different Gram-negative bacteria, transforming them into CRE [54]. The carbapenemase genes are frequently transmitted along with other antibiotic resistance genes. Therefore, the usage of other antibiotics can also result in selection pressure for the emergence of CRE, such as resistance against cephalosporins and fluoroquinolones [50].

### 6.2. Colonisation

CRE colonise the digestive tract, which serves as their reservoir. CRE colonisation later leads to severe infections such as pneumonia, followed by UTIs, septicaemias, cutaneous infections, and surgical site infections [55]. Long-term carriages in the intestine have been reported to be as long as 2 years without spontaneous clearance [56,57].

A recent systematic review reported the role of the hospital environments, such as wastewater drainage. It sinks to act as a reservoir of CRE. This study identified 32 CRE outbreaks associated with wastewater reservoirs of CRE such as *Enterobacterales*, *Pseudomonas,* and *Acinetobacter* [58]. Several other studies that employed whole genome sequencing (WGS) methods have also substantiated the presence of bacteria that carry the carbapenemase genes on their plasmids and confer resistance to other bacteria, especially to the bacteria of *Enterobacterales* species, conferring resistance against carbapenem [59,60,61]. These observations warrant high throughput genomic screening/surveillance to trace the bacteria carrying carbapenem-resistance plasmids.

Research has narrowed the predisposing factors for a patient contracting CRE to the following: overnight stay in a medical facility; chemotherapy; previous history of CRE infection; and epidemiological contact with a known carrier—all within one year [62]. Other identified CRE carriers are hematopoietic stem cell transplant recipients, and newborns, particularly when they have a history of carbapenem treatment [63,64].

Strict measures should be taken to maintain a closed sterile environment for the hospitalised patients and adequate steps should be taken to avoid contamination. Different CRE infection control measures proposed by the ECDC are screening for rectal CRE carriage on hospital admission, hand hygiene, patient isolation, antibiotic restriction, awareness, and education imparted to the patients and the staff [65]. A recent study from Paris recommended pre-emptive isolation, screening all patients who have been under in-patient care in other countries within the past 12 months, contact hygiene, and cohorting patients in separate areas as per carriage/contact status [66]. The WHO has also outlined similar guidelines for preventing and controlling CRE infections [67].

Antimicrobial stewardship should also be followed along with infection control measures to control or prevent the rise in CRE. Reduced carbapenem use has been shown to be effective for CRE control [47].

### 6.3. Transmission of CRE to the Community

The CRE have excellent potential to spread in the community. The CRE can be transferred to humans through the food chain (animals as a food source). The exogenous pathogenic *E. coli* are reported to be transmitted through animal sources to humans [68]. Feco-oral transmission and accumulation along food chains have also been reported to aid in spreading carbapenem-resistant *E. coli* in the healthy population. Such patients carry the CRE in the intestinal flora and serve as potential candidates to spread CRE to other patients. Several studies have reported the contamination of food items such as chicken, meat, and vegetables [69,70,71,72,73,74]. CRE have also been detected in poultry, pet birds, dogs, pigs, cattle, horses, seafood, cats, swallows, wild stork, wild boars, black kites, and gulls [75,76,77,78].

For communities, CRE acquisition is either through horizontal transmissions or carbapenem-susceptible strain-associated resistance [79]. The lack of widely varied core genome diversity exhibited by epidemiological findings explains that patient-to-patient transmission is of significant concern. In control of CRE, dissemination is aimed at improving hand hygiene, early detection of the carriage, cohorting patients with committed staff, instituting contact precautions, and improving environmental cleaning [80]. Another approach recommended is decolonisation protocols, but it lacks sufficient evidence in its support [79,80]. Hand hygiene improvement is the primary intervention that can be used to prevent and control CRE dissemination. In an ICU environment, mathematical models have been used successfully in the prediction of contact precautions, and hand hygiene is an effective intervention for the control of CRE spread [79,81,82]. According to the authors of a recent study, achieving hand hygiene compliance as a strategy for CRE control reduces CRE colonisation rates. Still, it does not completely eliminate CRE because colonised patients continue to be admitted to the ICU due to infections from other wards such as the emergency room (ER) [83]. The findings from these studies explain the value of implementing CRE prevention strategies such as hand hygiene and contact precautions in all the units of a hospital to reduce the prevalence rate for the infection spread in the ICU.

A need to evaluate the interactions between the community and hospital settings on the acquisition of CRE is an approach that is proposed to enhance infection control. In a recent study, the author believes that the emergence of carbapenem-resistance strains is also common in agricultural and environmental settings [84]. CRE was isolated from sewage from hospitals in Brazil, Spain, and China as well as in community sources of water in Brazil, Italy, and China [79]. Recently, findings have revealed the presence of CRE in fresh vegetables and spices in Asia and retail chicken meat in Egypt. However, these findings identify that dissemination of CRE still remains unclear for most studies [79,82,83]. Interaction between the development of community-associated infection (CAI) and CRE in the environment is not yet established [85]. CRE identified in patients who are admitted to hospitals are considered a community-onset infection (COI) as a particular patient has a multiple exposure history within a healthcare system either in outpatient or inpatient departments [86]. Infection classification can be considered in CAI if the patient presents a history of exposure previously to a healthcare system. Therefore, it is challenging to differentiate COI and CAI, but reports indicate that a third (30%) of CRE patients on hospital admission did not present a previous history of exposure within a healthcare system [85,87]. CAI-associated CRE is linked to different settings worldwide with significant variations in incidence rate, with CAI or COI-CRE proportions estimated to range between 0 and 29% [79]. Though most incidences are characterised as low due to effective CRE control, a patient could be colonised by CRE asymptomatically and become the source of multi-drug resistant bacteria in a healthcare facility.

### 6.4. Screening for Carbapenem Resistance

Rapid detection and isolation of carriers on admission has been shown to significantly reduce the rates of hospital transmission of CRE infections [79]. The same study identified that information sharing for the CRE colonised patients within the government databases could also help to reduce the carriers of CRE by over 20%. Additionally, COI-CRE patients are much easily identified in comparison to those with CAI [86]. COI patients can be screened by identification of known risk factors such as exposure to antimicrobial use and the healthcare system. Even with patients with no previous diagnosis of CRE colonisation, there is a need to conduct adequate screening and placing the patients on contact precaution for CRE. Long-term care facilities are a well-described source of the CRE carriage, with the incidences of the infection in these settings being much higher than other acute care hospitals [79,83,84]. Recent travel to areas considered to have the endemic of the infection could be another concern for screening.

### 6.5. Phenotypic and Genotypic Characterisation of CRE

Development and implementation of control and management options for CRE infection greatly rely on efficient and reliable laboratory detection and identification of CRE. However, the detection of CRE becomes challenging because of the heterogeneous expression of the CRE resistance genes and the multitude of pathways/mechanisms involved in conferring resistance to the bacteria [87]. Different mechanisms by which CRE acquire resistance to carbapenemase are the structural changes in the porins that limit the access of the drugs to their targets, changes in the binding sites for penicillin, overactivation/overexpression of efflux pumps, and production of different carbapenemases [88,89]. Resistance can be gained by a single mechanism or by the interplay of more than one mechanism. In recent years, there has been great progress in microbiological and molecular biology techniques to detect and characterise pathogens, especially the Gram-negative carbapenemase-resistant bacteria. These methods can be broadly classified as phenotype-based methods and DNA-based methods. The detection methods that reply on the phenotypic characters of the microbes for identification are carbapenemase enzyme activity assays, immunochromatographic (IC) assays, colorimetric methods, and the DNA bases/molecular methods, those that exploit the DNA sequencing methods for the detection such as polymerase chain reaction (PCR) and the WGS [90]. The comparison of conventional methods for the detection of antimicrobial patterns among bacterial isolates versus genotypic methods is shown in Figure 3.

### 6.6. Phenotypic Characterisation

#### 6.6.1. Antibiotic Susceptibility Test

This is among the most typical and first-line methods for detecting carbapenemases. Several indicators could be utilised for this, but ertapenem is generally considered the most sensitive. MIC breakpoints are also used to confirm the sensitivity tests. These testing methods show high variability in detecting carbapenem resistance. The breakpoints suggested by the European Committee on Antimicrobial Susceptibility Testing (EUCAST) are helpful in predicting carbapenemase-resistant Gram-negative bacteria, but by this method, some bacteria remain susceptible to many carbapenems. For instance, some *Enterobacterales* isolates carry the antibiotic resistance gene *blaKPC* but can be acted upon by ertapenem, meropenem, and imipenem [81,91,92]. The Clinical and Laboratory Standards Institute (CLSI) has formulated recommendations and guidelines for detecting CREs based on MIC breakpoints [93]. KPC is much more sensitive to this method as compared to NDM and IMP. As per CLSI, the carbapenemase-producing organisms should be resistant to all third-generation cephalosporins, but some CRE-producing OXA-48 are sensitive to third-generation cephalosporins [86]. The CLSI also suggested that when *Enterobacterales* isolates are suspected to be producing carbapenemase by carbapenem breakpoints, they should be further confirmed by molecular techniques and others such as MHT and the Carba NP test [94]. Although these are the first line of detection techniques for carbapenemase-producing organisms, they too fail to detect most of carbapenemase-producing microorganisms [95]. The advantages and disadvantages of certain phenotypic tests for the detection of carbapenemase producing CRE clinical isolates are shown in Figure 4.

#### 6.6.2. Multi-Disk Mechanism Testing and Combined Disk Synergy Tests

These methods rely on the inhibitors such as boronic acids for KPC detection, Ethylenediaminetetraacetic acid (EDTA) or dipicolinic acid for MBL, avibactam (NXL104) for OXA-48, clavulanate for ESBL, and cloxacillin for *AmpC* [96]. These methods are popular because they are simple and inexpensive. These tests have variable sensitivities (90% to 100%) [97].

The Mastdiscs ID inhibitor combination detection disks (MDI) method detects carbapenemase by calculating the inhibition zones of the disks containing the enzyme inhibitors. Reports suggest that this method is insufficient to differentiate between OXA-48-type genes and different MBL genes such as NDM, VIM, and IMP [98]. The Rosco Diagnostica Neo-Sensitabs (RDS) is a carbapenemase detection method. Both methods (MDI and RDS) are not efficient enough to detect OXA-48-like enzymes [99]. The OXA-48 disk test can differentiate OXA-48-producing CRE from others [100].

#### 6.6.3. Chromogenic Media

These methods are considered one of the optimal methods to screen for CRE [101]. They are based on the principle of the interaction of a chromogenic enzyme substrate and the specific resistance enzymes produced by the CRE. These enzymes catalyse the substrate, and the CRE is identified based on the colour of the product formed. The sensitivity of this method ranges from 13% to 100% [102]. These methods have a low sensitivity for detecting OXA-48-producing CRE. Recently a new screening medium (Supercarba) has been reported to have higher sensitivity (96.5%) and can detect all the carbapenemase, including OXA-48 [103]. The biggest drawback of these media is that they detect the carbapenemase in lactose-fermenting bacteria. These media have a short shelf-life ranging from 7 to 10 days [104]. The modified version of Supercarba medium (trade name: mSuperCarba) with a shelf-life of up to one month is 100% sensitive and specific for KPC, MBL, and OXA-48-type of CRE [105]. These methods are advantageous because of the short turnaround times of detection.

#### 6.6.4. Modified Hodge Test (MHT)

The CLSI for many years suggested the MHT. This growth-based approach was widely used to detect carbapenemase producers, until 2018, when it was removed from the CLSI M100 owing to various drawbacks and better alternatives. The MHT is a good assay in general, although it has very low sensitivity for MBLs and excellent sensitivity for class A carbapenemases (especially KPC) and class D carbapenemases [6]. This test detects carbapenemase enzymes in the *Enterobacterales* isolates. MHT is used to detect resistance by quantifying the concentration of carbapenem, which is broken down by carbapenemase produced by resistant organisms, allowing carbapenem-susceptible *Escherichia coli* to survive. It is sensitive to detecting VIM, IMP, and OXA-48 [106]. The major problem with these tests is the higher rates of false positives and false negatives when detecting some AmpC-producing bacteria harbouring mutations in porin. Despite the disadvantages, this method is widely suited for large-scale screening of carbapenemase because of its cost-effectiveness [107]. The turnaround time is between 18 and 24 h. Recent studies reveal that 57% and 84% of labs still employ the MHT as their carbapenemase detection method in Europe and California, respectively, even though the CLSI and EUCAST no longer recommend it. It is being used in many laboratories [106].

#### 6.6.5. Carba NP

The Carba NP employs colorimetric analysis for the rapid detection (≤2 h) of CPE. It is one of the standard phenotypic techniques recommended by the CLSI and was first presented in 2012 as a rapid test for carbapenemase identification in both *Enterobacterales* and *Pseudomonas* spp. Imipenem is hydrolysed by carbapenemase, causing changes in pH, which are detected by colour changes in phenol red, a pH indicator. This technique is fast, cost-effective, sensitive, and specific as compared to several other phenotypic methods [108]. Carba NP is considered a confirmatory test for CRE by the CLSI and EUCAST. The sensitivity range of this test is 73–100%, but it has a low sensitivity for the detection of OXA-48 and some class A carbapenemase such as GES-5 and SME-1 as they have low imipenem hydrolysis activity [109]. Therefore, if OXA-48 is suspected or the isolates are mucoid, this procedure should not be used. The cost of Carba NP is reasonable and ranges from USD 2 for manual versions to USD 15 for specific commercial versions. If the short shelf-life reagents are not utilised within three days owing to intrinsic imipenem instability in the solution, the cost may rise further. Similar to the MHT, visual interpretation of the outcomes may sometimes be subjective due to how slightly the colour shift might vary. Mucoid isolates have been reported to be more challenging to recognise and often provide a large percentage of false negative results. As a result of its fastest turnaround time and better clinical outcomes, the Carba NP test is a suitable approach for carbapenemase identification [108]. 

#### 6.6.6. Carbapenem Inactivation Method (CIM) and Modified CIM

This method was first described in 2015. CIM detects the rate of carbapenem inactivation by the carbapenemase-producing organisms. Bacterial colonies are incubated in water suspended with carbapenem disks (usually a 10 μg meropenem) and incubated. Then the discs are placed in carbapenem-sensitive E. coli strain plates with Mueller–Hinton agar (MHA). A small zone of inhibition is indicative of a carbapenemase-producing organism. This method of detection of CRE is widely utilised due to its specificity, sensitivity, and cost-effectiveness. Studies have shown CIM to be more accurate than other phenotypic tests, such as Carba NP and MHT, for detecting CRE. CIM also detects OXA-48 with high sensitivity [110]. This test is famous because of its simplicity and the non-requirement of special equipment. This method has demonstrated excellent sensitivity (91–94%) and specificity (99–100%). Some studies have shown that the CIM performs better than the Carba NP test, particularly when it comes to identifying class D carbapenemases. The CIM has several benefits, including minimal cost (i.e., less than USD 1), the convenience of use, and objective interpretation. It only requires the basic chemicals and medium used in microbiology labs (as opposed to carba NP). Similar to the MHT, the fact that the CIM’s overnight incubation before findings are available is one of its significant downsides. Even though the CIM has shown outstanding results, several investigations have found that *OXA-48*-like carbapenemases in the *Enterobacterales* have limited sensitivity [109,110]. Due to this, a CLSI working group developed a modified carbapenem inactivation technique (mCIM), a version of the CIM, which indicated an improvement in sensitivity from 82% to 93% while retaining 100%. The mCIM generally performed well in detecting class A, B, and D carbapenemases. The mCIM has a longer incubation period of 4 h than 2 h and prepares the bacterial suspension using tryptic soy broth (TSB) rather than water. In 2017, the CLSI agreed to include the mCIM as an authorised and validated technique in the CLSI M100 publication. To identify MBLs from serine carbapenemases in *Enterobacterales*, another variation of the CIM, known as the eCIM, was added to the CLSI recommendations in 2018. It should be used in combination with the mCIM [110].

#### 6.6.7. Immunochromatographic (IC) Assays

Numerous IC assays rely on antigen–antibody interaction to detect carbapenemase antigens. Commercial kits (KPC-k-set and OXA-48 K-set tests) combine nanotechnology principles and antigen–antibody interaction to detect OXA-48-like enzymes. These kits use gold nanoparticles cross-linked to nitrocellulose membranes [111]. These test kits have high accuracy of detection. The OXA-48 K-set can identify all OXA-48-producing organisms and be 100% sensitive and specific. These kits have a very short time of detection (10 min). This kit can distinguish between the allelic variants of OXA-48 such as OXA-204, OXA-244, OXA-181, and OXA-232 [111]. These kits can detect carbapenemase producers directly in clinical specimens such as blood cultures and rectal swabs [103].

#### 6.6.8. Bio-Analytical Methods—MALDI-TOF MS

Matrix-assisted laser desorption/ionisation-time of flight mass spectrometry (MALDI-TOF MS) is based on the analytical detection of various chemical products of the pathogens based on their molecular weight. It is now used to detect β-lactamase activity to detect Gram-negative bacteria [112]. The protein extracts of bacterial cultures are incubated with carbapenems, and the degradation products of the β-lactams are measured by MALDI-TOF MS. This method has a sensitivity of around 97%. Still, it has been shown to produce false positive and false negative results [113]. In addition to the difficulty in detecting OXA-48, and false negative results, this technique is costly and requires expertise, and hence is less popular in laboratory settings.

### 6.7. Genotypic Characterisation of CRE: Detection of Carbapenemase Genes

DNA-based techniques are more reliable and accurate than other techniques to detect CRE [114]. DNA-based techniques are the most widely used techniques that exploit the uniqueness of the DNA sequence of the organisms to identify them. Polymerase chain reaction (PCR)-based techniques, hybridisation-based techniques, and whole genome sequencing (WGS) are the molecular detection methods used to identify CRE. The basis of molecular approaches is the identification of carbapenemase genetic determinants, either from bacterial isolates or directly from patient samples. Conventional or real-time PCR is the most common method for using nucleic acid amplification to identify carbapenemases. Industrial manufacturers have developed various molecular techniques during the past several years, which has boosted their use in healthcare facilities. Due to their high costs, clinical labs do not use these techniques extensively or frequently. Even though molecular diagnostics is the most common way to diagnose CPE, not all carbapenemases and/or variants can be found. Commercial panels are often made to find only the most common enzymes [93].

The first molecular test that received FDA approval was the BioFire Filmarray^®^ Blood Culture Identification Panel from BioMerieux. An automated multiplex PCR technique called Filmarray^®^ can quickly identify 24 pathogens from positive blood cultures, including three resistance mechanisms (*mecA, vanA/B*, and *blaKPC*) and 8 Gram-positive, 11 Gram-negative, and 5 yeast species [113].

The Nanosphere Verigene^®^ Gram-negative Blood Culture Test is another FDA-cleared panel that can identify the ESBL CTX-M and five common carbapenemases (KPC, VIM, NDM, IMP, and OXA) from positive blood cultures in as few as two hours. The Verigene^®^ test has shown outstanding efficacy in multicentre investigations for identifying carbapenemase genes [113].

The KPC, NDM, VIM, IMP, and OXA-48 families of carbapenemase-producing organisms have 91 gene targets that may be detected using the real-time PCR test Xpert^®^ Carba-R (Cepheid), which is also an FDA-approved method. This test offers the benefit of assessing samples from resistant cultures directly from rectal swab specimens, blood, urine or sputum in just 48 min [113]. Several different molecular tests have been used in research facilities to find carbapenemase genes, but they still need further validation before being used regularly in clinical settings. Even though molecular-based approaches are the gold standard for carbapenemase detection, they should not be used for normal clinical use because they are too expensive. Instead, they should be used for infection control, research, and complex patients who may benefit based on clinical assessment [112].

#### 6.7.1. Polymerase Chain Reaction (PCR)

PCR-based methods amplify the signature sequences present on the chromosomal DNA of the bacteria for their identification. PCR-based methods are employed to confirm the results of phenotype-based methods. PCR can be performed on the DNA isolated from bacterial cultures or directly from clinical samples (real-time PCR). Numerous PCR-based detection methods have been developed to identify CRE [115]. Many multiplex PCR methods can identify more than one bacterium at one time. Identification of 11 carbapenemase genes in 3 different multiplex reactions has been reported. The genes detected by this multiplex method were VIM, IMP, KPC, NDM, SPM, AIM, SIM, OXA-48, GIM, DIM, and BIC. This method has shown promising results in identifying many clinical strains of carbapenemase producers, especially Gram-negative bacteria such as *K. pneumoniae, E. coli, Citrobacter freundii, P. aeruginosa, Enterobacter cloacae,* and *A. baumannii* [115].

By amplifying and tracking the bacterial DNA sequences in real-time, multiplex real-time PCR identifies the bacterium. Due to the quick turnaround times of detection, which can be as quick as an hour, it has grown in popularity in recent years. The sensitivity range for PCR-based techniques is between 97 and 100% [116]. These PCR-based techniques demand specialised knowledge and pricey equipment despite being susceptible and fast. Additionally, these techniques are useless for locating novel CRE strains.

#### 6.7.2. Microarrays

The principle behind microarray is the hybridisation of the target sample with the probes. Multiple samples are detected by microarray in a single reaction. The probes on the chip hybridise with the bacterial nucleotide sequences before the chip is checked for hybridisation. Numerous bacterial species have been reported to be easy to identify using microarrays [117]. Multiple carbapenemase genes have been discovered using microarray. Microarray has proven to be incredibly sensitive and specific in several investigations [118]. This approach for identifying infections has very high throughput but is labour-intensive and expensive.

#### 6.7.3. Whole Genome Sequencing (WGS)

To analyse the complete DNA sequence of the bacteria, WGS uses the next-generation sequencing (NGS) method. In a short time, WGS analyses the entire bacterial genome. Numerous epidemiological research has made use of it [119,120]. WGS, however, is pricy and necessitates specialised equipment and knowledge. Consequently, it is not frequently employed in therapeutic settings.

## 7. Conclusions

Antimicrobials are used more frequently due to improvements in medication discovery and delivery. Due to selection pressure, AMR/MDR bacteria have emerged due to the unprecedented use of antibiotics. Public authorities should also raise knowledge regarding the development of AMR and MDR among medical professionals and the local population. Only after a licensed doctor has prescribed a legitimate prescription should antibiotics be given out. Since CRE can vertically transfer the genetic materials carrying the resistance genes to other bacteria, hospitals should take the necessary precautions to guarantee effective containment and disinfection of the potential sources of contamination or infection. Even though there are several genotypic and phenotypic screening methods for CRE, there is still lack of tests or methods for screening all the many carbapenemase-producing species. The robust techniques to identify the CRE efficiently with high sensitivity are expensive. Therefore, there is a need to develop a cost-effective and sensitive technique to detect and identify all the different types of carbapenemase-producing organisms, especially the CRE.

Although the World Health Organization has identified CRE as a priority pathogen for prioritising new drug development and initiated the development of new drugs such as zidebactam, taniborbactam, nacubactam, etc., it is mandatory to control the use of drugs at the end-user level. There should be antibiotic stewardship programs at the institutional level that monitor the use of antibiotics. The proper and timely use of available antibiotics may help to stop the emergence of high AMR rates.

## Figures and Tables

**Figure 1 medicina-58-01675-f001:**
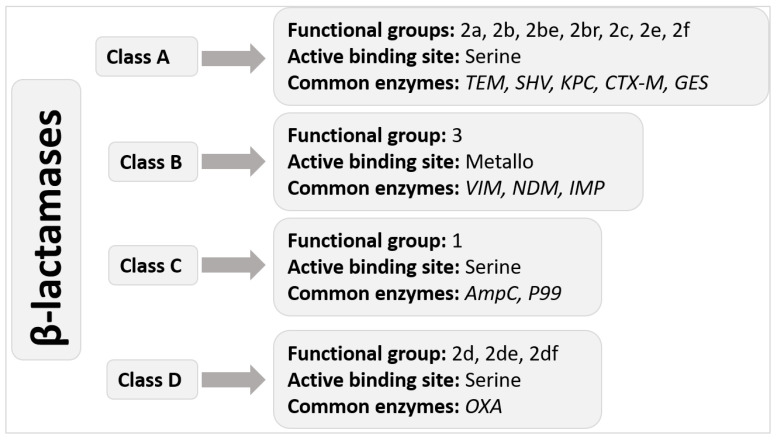
Ambler molecular classification of beta-lactamases.

**Figure 2 medicina-58-01675-f002:**
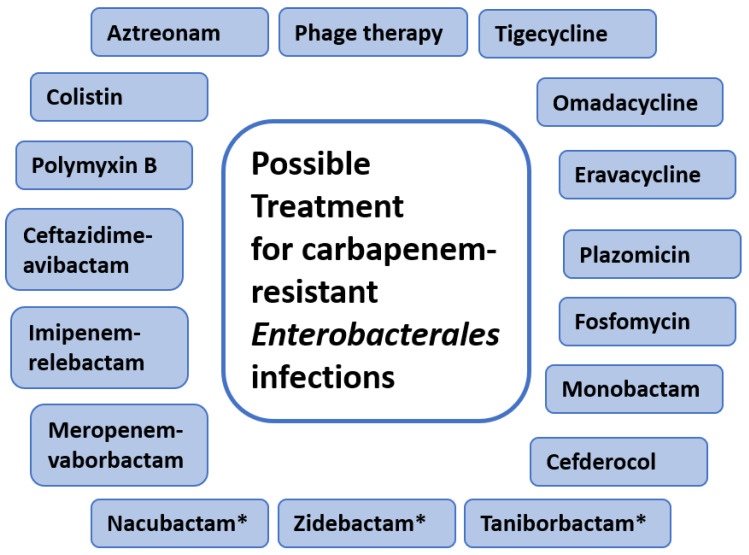
Possible therapeutic options for treatment of infection by CRE. * Represents the antibiotics that are currently under development processes or under clinical trials.

**Figure 3 medicina-58-01675-f003:**
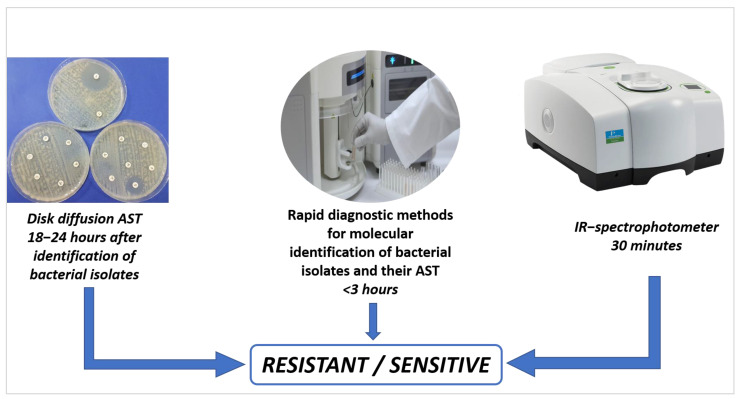
Gold standards method for AST (disc diffusion) vs. automation methods.

**Figure 4 medicina-58-01675-f004:**
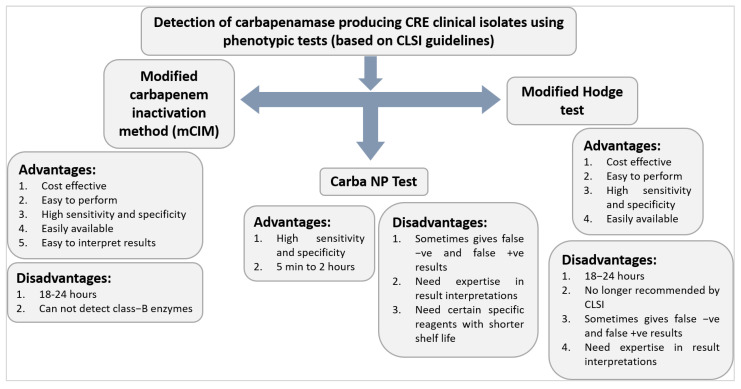
Characteristics of certain phenotypic tests (based on CLSI guidelines) for the detection of carbapenemase-producing CRE clinical isolates.

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
