# Peer review of "An Overview on Phenotypic and Genotypic Characterisation of Carbapenem-Resistant Enterobacterales"

_medicina, 2022, doi:10.3390/medicina58111675_

Round 1

Reviewer 1 Report

Ali A. Rabaan (medicina-1994399) and colleagues presented an informative description of all the CRE in the field, however, the overall manuscript is to tell different parts without a clear connection among each part, additionally, the summary figures are usually needed for the review paper. Moreover, add the different approaches were presented and introduced, but their pros and cons should be well commented, at least, their economic efficiency, time efficiency, advantage and disadvantage should be there (comparative tables would be an ideal way to show this purpose). The insight clue for improving control of CRE is also needed, where is the future for addressing the problem. All these parts should be in the updated and revised review.

Considering the importance and hot topic right now, colistin could be an option for treating CRE infection, however, the reference (DOI: 10.3390/microorganisms7100461) and corsspond part for this topic should be improved.

Author Response

Reviewer 1

Comments and Suggestions for Authors

Ali A. Rabaan (medicina-1994399) and colleagues presented an informative description of all the CRE in the field, however, the overall manuscript is to tell different parts without a clear connection among each part, additionally, the summary figures are usually needed for the review paper. Moreover, add the different approaches were presented and introduced, but their pros and cons should be well commented, at least, their economic efficiency, time efficiency, advantage and disadvantage should be there (comparative tables would be an ideal way to show this purpose). The insight clue for improving control of CRE is also needed, where is the future for addressing the problem. All these parts should be in the updated and revised review.

Response: Dear reviewer, we really want to appreciate your kind struggles to review the current review article for possible publication. In order to address your comments, we have added 4 figures in the revised version of manuscript. Line 121-145, 401-405, 412-416, 418-421, 427-438, 447-464, 491-520: The methods have been discussed in detail in the revised version of manuscript. Furthermore, the conclusion section has been improved.

Considering the importance and hot topic right now, colistin could be an option for treating CRE infection, however, the reference (DOI: 10.3390/microorganisms7100461) and corsspond part for this topic should be improved.

Response: Dear reviewer, thank you for your suggestion. We agree that the review on colistin resistance among bacterial isolates will give a better readership to the article. But the aims of this current review were to focus on CRE first. We are planning other review articles on different aspects of AMR. In future articles, our plan is to give a thorough review of specifical drugs like colistin. However, we can compile all information in one article too, but it will increase the length of article and then might be led to less information. Hence, we request you to let us proceed with CRE in the current review literature.

Reviewer 2 Report

Dear author,

The manuscript "A review on treatment modalities and genotypic-phenotypic characterization of Carbapenem-resistant Enterobacterales" offers a good and complex image of the carbapenemase-production species. Even though there are several genotypic and phenotypic screening methods for CRE, there is still lack of tests or methods for screening all the many carbapenemase-producing species. 

I have some suggestions:

1. Please add kre recent breferences.

2. Please recheck your english.There are some mistakes in spelling correct some words.

3. line 457 please corect : carbapanamase.

Thank you!

Regards!

Author Response

Reviewer 2

Comments and Suggestions for Authors

Dear author,

The manuscript "A review on treatment modalities and genotypic-phenotypic characterization of Carbapenem-resistant Enterobacterales" offers a good and complex image of the carbapenemase-production species. Even though there are several genotypic and phenotypic screening methods for CRE, there is still lack of tests or methods for screening all the many carbapenemase-producing species. 

Response: Dear reviewer, we really want to appreciate your kind struggles to review the current review article for possible publication. Line 121-145, 401-405, 412-416, 418-421, 427-438, 447-464, 491-520: The methods have been discussed in detail in the revised version of manuscript.

I have some suggestions:

  1. Please add kre recent breferences.

Response: Dear reviewer, the reference section has been rechecked. Majority of the cited references are after 2010.

  1. Please recheck your english. There are some mistakes in spelling correct some words.

Response: The manuscript has been thoroughly revised for English proofreading and grammatical mistakes.

  1. line 457 please corect : carbapanamase.

Response: The word has been corrected throughout the manuscript and highlighted in red colour.

Thank you!

Regards!

Reviewer 3 Report

The review article is containing superficial knowledge with limited information in different areas. The review seems to be focusing on diagnostic techniques. The topic seems to be very attractive, but contents are not according to the topic. Molecular aspects are limited which need to be explained well. Tables or figures can be added based on available data. If authors invest more time to explore in detail about every aspect by adding more data, the manuscript will get published in high impact journal.

Manuscript in its current state can not be accepted for publication.

Author Response

Reviewer 3

Comments and Suggestions for Authors

The review article is containing superficial knowledge with limited information in different areas. The review seems to be focusing on diagnostic techniques. The topic seems to be very attractive, but contents are not according to the topic. Molecular aspects are limited which need to be explained well. Tables or figures can be added based on available data. If authors invest more time to explore in detail about every aspect by adding more data, the manuscript will get published in high impact journal.

Response: Dear reviewer, we really want to appreciate your kind struggles to review the current review article for possible publication. In order to address your comments, we have added 4 figures in the revised version of manuscript. Line 121-145, 401-405, 412-416, 418-421, 427-438, 447-464, 491-520: The methods have been discussed in detail in the revised version of manuscript. Furthermore, the conclusion section has been improved.

Manuscript in its current state can not be accepted for publication.

Response: Dear reviewer, we hope that after addressing your comments, the manuscript is more suitable for further steps towards publication. If still it requires any addition or changes, we would love to address more comments, as the purpose is to make it good for the reader and scientific community.

Round 2

Reviewer 1 Report

The improved manuscript now are more appropriate, and a few related papers in the filed should be quoted.

1. doi: 10.1016/j.micres.2022.127249

2. doi: 10.3389/fmed.2020.616490.

3. doi: 10.1016/j.jhin.2022.10.011.

4. doi: 10.14740/jocmr4764. Epub 2022 Jul 29.

5. doi: 10.3390/microorganisms7100461.

Author Response

Reviewer 1

Comments and Suggestions for Authors

The improved manuscript now are more appropriate, and a few related papers in the filed should be quoted.

  1. doi: 10.1016/j.micres.2022.127249
  2. doi: 10.3389/fmed.2020.616490.
  3. doi: 10.1016/j.jhin.2022.10.011.
  4. doi: 10.14740/jocmr4764. Epub 2022 Jul 29.
  5. doi: 10.3390/microorganisms7100461.

Response: Dear reviewer, thank you for your valuable suggestions. Your suggested paper has helped us to modify the writeup in a better way. Hence we took the information and cited them as ref # 7, 8, 13, 53 and 84.

Reviewer 3 Report

Title should be revised as

An overview on phenotypic and genotypic characterisation of Carbapenem-resistant Enterobacterales

Author Response

Reviewer 3

Comments and Suggestions for Authors

Title should be revised as

An overview on phenotypic and genotypic characterisation of Carbapenem-resistant Enterobacterales

Response: Dear reviewer, thank you for your valuable suggestion. The title of manuscript has been amended in the revised version of manuscript.